# SPECIALIZED TRANSFORMERS: FASTER, SMALLER AND MORE ACCURATE NLP MODELS

## ABSTRACT

Transformers have greatly advanced the state-of-the-art in Natural Language Processing (NLP) in recent years, but are especially demanding in terms of their computation and storage requirements. Transformers are first pre-trained on a large dataset, and subsequently fine-tuned for different downstream tasks. We observe that this design process leads to models that are not only over-parameterized for downstream tasks, but also contain elements that adversely impact accuracy of the downstream tasks. We propose a Specialization framework to create optimized transformer models for a given downstream task. Our framework systematically uses accuracy-driven pruning, i.e., it identifies and prunes parts of the fine-tuned Transformer that hinder performance on the downstream task. We also replace the dense soft-attention in selected layers with sparse hard-attention to help the model focus on the relevant parts of the input. In effect, our framework leads to models that are *not only faster and smaller, but also more accurate*. The large number of parameters contained in Transformers presents a challenge in the form of a large pruning design space. Further, the traditional iterative prune-retrain approach is not applicable to Transformers, since the fine-tuning data is often very small and re-training quickly leads to overfitting. To address these challenges, we propose a hierarchical, re-training-free pruning method with model- and task-specific heuristics. Our experiments on GLUE and SQUAD show that Specialized models are consistently more accurate (by up to 4.5%), while also being up to $2.5\times$ faster and up to $3.2\times$ smaller than the conventional fine-tuned models. In addition, we demonstrate that Specialization can be combined with previous efforts such as distillation or quantization to achieve further benefits. For example, Specialized Q8BERT and DistilBERT models exceed the performance of BERT-Base, while being up to $3.7\times$ faster and up to $12.1\times$ smaller.

## 1 INTRODUCTION

Transformers models such as BERT (Devlin et al., 2019), GPT-2 (Radford et al., 2019) and GPT-3 (Brown et al., 2020) have revolutionized the field of Natural Language Processing (NLP), greatly advancing the state-of-the-art in many NLP tasks. Models that achieve good performance on these tasks are of high practical significance, finding their place in commercial applications such as social media monitoring (sentiment analysis), AI chat assistants (question answering), automated summarization tools (analyzing sentence similarity), *etc*. Therefore, there is a strong interest in creating more accurate and efficient Transformer models for these tasks.

Transformers are first pre-trained on very large datasets, and subsequently fine-tuned for different downstream tasks. However, the current method of pre-training and fine-tuning Transformers has two major drawbacks. First, fine-tuning the pre-trained Transformers leads to models that are highly over-parameterized models for the downstream tasks, especially since many of these tasks have very limited training data. This could lead to unstable models (Dodge et al., 2020) with sub-optimal generalization ability (Michel et al., 2019). Second, these large fine-tuned models present high computation and storage requirements for inference. This problem is further exacerbated by the trend towards larger and more accurate models over time. For instance, increasing the number of parameters from 1.5B to 175B enabled a reduction in perplexity for Language Modelling (on the Penn Treebank dataset) from 35.8 in GPT-2 (2019) to 20.5 in GPT-3 (2020). In this work, we address both these challenges, taking advantage of the over-parameterized nature of pre-trained models to

create individually Specialized models for the different downstream tasks that are smaller, faster and more accurate than the conventional fine-tuned models.

Prior research efforts have explored approximation techniques, such as quantization, pruning and distillation, for improving the inference efficiency of various classes of neural networks. However, these techniques invariably involve a trade-off between accuracy and efficiency. In addition, the vast majority of these techniques require re-training or additional fine-tuning. This becomes especially problematic for Transformers, since these large models require significant time and energy to train and fine-tune. Fine-tuning is usually performed on a very small data set, and as a result, conventional iterative prune-and-retrain methods quickly end up overfitting when applied to Transformer models. In contrast, our Specialization framework utilizes the unique characteristics of Transformer training and deployment to enable substantial gain in both accuracy as well as efficiency. Our framework also does not require any additional re-training or fine-tuning, and can be applied in a plug-and-play manner to any Transformer model that is fine-tuned for any downstream task.

During pre-training, Transformers capture rich linguistic knowledge and gain a deep understanding of the structure of the target language. Fine-tuning refines this knowledge for a specific downstream task by training a task-specific final layer. We observe that, due to the nature of their design process, Transformer models are not only over-parameterized but also contain parts that are, in fact, harmful to performance of downstream tasks. In order to exploit this observation in a principled manner, we introduce a framework to identify and prune the harmful elements of a Transformer (parameters, grouped at different levels of granularity *i.e.*, self-attention blocks, feed-forward neural network blocks, attention heads and neurons), with the goal of maximizing accuracy on the downstream task. In contrast with prior pruning methods that prune elements with little-or-no impact on the network output, the proposed method prunes elements that have a considerable impact on the output, leading to the highest positive impact on accuracy.

In order to reduce the large pruning space, we analyze the different elements of the fine-tuned Transformer in a hierarchical manner, starting with entire self-attention or feed-forward neural network blocks, followed by attention heads, and neurons, and prune the harmful elements. The core of the Transformer is self-attention, where each token in the input builds its representation based on the extent of attention it places on all the other tokens. However, we observe that in some cases, restricting the attention span of each token to only focus on the relevant tokens (in certain layers) leads to better information flow inside the model. Hence, our framework is also equipped with the ability to identify the appropriate layers and replace the "soft" self-attention with "hard" self-attention in these layers. We also introduce Transformer-specific heuristics to minimize the run-time of our framework, thereby enabling it to scale to large Transformer models.

We summarize our main contributions as follows:

- We introduce a Specialization framework that optimizes Transformer models for specific downstream tasks through the use of accuracy-driven pruning and selective hard attention.
- We incorporate multiple heuristics in the framework, such as hierarchical processing, model-driven insights, and run-time based ordering of elements, in order to minimize the overheads.
- We propose a significance analysis technique to identify the importance of each element of the fine-tuned Transformer for a given downstream task. We use this technique to prune elements that are harmful to performance on the downstream task.
- We propose the selective replacement of the "soft" self-attention with hard attention in the appropriate layers, helping the model focus only on the relevant parts on the input to build better representations.
- Across a suite of different Transformer networks, we demonstrate that Specialized models are consistently more accurate and stable, while also being significantly faster and smaller than their conventional fine-tuned non-Specialized counterparts.

## 2 RELATED WORK

**Task-agnostic optimizations.** Given the effectiveness and popularity of Transformer models, several techniques have been proposed to overcome their computational and memory challenges, and

to accelerate inference using these models. A vast majority of these works introduce task-agnostic optimizations, using popular approximation techniques such as knowledge distillation (Sanh et al., 2019; Sun et al., 2020; Wang et al., 2020), early exit/ depth modulation (Elbayad et al., 2020; Xin et al., 2020; Zhou et al., 2020), quantization (Zafrir et al., 2019), attention head pruning (Zhang et al., 2020) and parameter sharing (Lan et al., 2020). In addition, Fan et al. (2020) randomly drop layers during pre-training, thereby enabling their dropping during inference; Khetan & Karnin (2020) learn the optimal sizes of the BERT elements during pre-training, and Wu et al. (2020) use Long-Short Range Attention to speed up the self-attention operation. Using DistilBERT and Q8BERT as examples, we demonstrate that our techniques are complementary to these works, and can be applied while fine-tuning for a specific task to create accurate ultra-efficient models for specific downstream tasks.

**Task-specific optimizations.** Hou et al. (2020), Shen et al. (2020), Jiao et al. (2019), Wang et al. (2020), Lagunas et al. (2021) and Sajjad et al. (2020) introduce task-specific optimizations, but the gain in efficiency comes at the cost of degradation in accuracy on the downstream task. In contrast, our framework improves both accuracy as well as efficiency, and hence appeals to a wider range of users. We also demonstrate that task-specific optimizations are more effective when applied to Specialized models compared to conventional fine-tuned models.

**Pruning techniques.** Finally, Structured Pruning has been applied to various classes of neural networks (Anwar et al., 2017; Molchanov et al., 2017), and greedy pruning strategies have also been explored to identify weights and parameters that the output is least sensitive to (Zhuang et al., 2018; Ye et al., 2020). In contrast, our method is designed to identify and prune parameters that have most detrimental effect on the output.

## 3 METHODOLOGY TO SPECIALIZE TRANSFORMERS

We propose a framework for producing Specialized Transformer models that are optimized for a specific downstream task, illustrated in Algorithm 1. Our framework performs two main optimizations: (1) It identifies and prunes elements that hinder performance on the downstream task at hand. (2) It selectively replaces soft self-attention with hard self-attention to help the model focus only on the relevant parts of the input.

### 3.1 ACCURACY-DRIVEN PRUNING

The problem of identifying an optimal set of elements to prune is challenging, and this is especially true for Transformers. In order to optimize a given model, we would ideally want to characterize the significance of each and every parameter in the model, rank them in order of importance, and finally prune only the least significant parameters. However, Transformers have billions of parameters, making this process computationally infeasible. In addition, previously proposed techniques that can efficiently estimate the importance of each parameter, such as using Taylor expansion, are not applicable. This is because the {approximate, fine-tune, approximate} cycle does not work for Transformers during fine-tuning, since they very quickly overfit the limited training data for the downstream tasks (usually within 5 epochs). We address both these issues through the use of a hierarchical greedy algorithm that does not require any additional training or fine-tuning. To determine the significance of each Transformer element, we first fine-tune the original Transformer model for the given downstream task to obtain the baseline loss. Then, for the element under consideration in each iteration of the framework, we compute the loss of the current Transformer model with the element removed. We prune the element under consideration if the validation loss when it is removed is less than the minimum loss seen thus far during the optimization process, since the goal is to find a model with minimum loss. Also, in order to prevent overfitting to the validation set, we introduce a generalization constraint in addition to the aforementioned loss condition. This constraint ensures that an element is pruned only if it decreases (or retains) the loss of at least a certain number ($N$) of samples in the validation set over the current best solution (computed by $num\_samples\_helped$ function in Alg. 1), where N is the number of elements in the validation dataset whose loss is less than the average loss of the misclassified samples (we consider samples whose loss is greater than the average loss of the misclassified samples to be outliers). Therefore, elements are pruned only if a vast majority of the samples in the validation set benefit from their removal, resulting in improved generalization performance. If the loss with the element removed is greater than the minimum loss

---

**Algorithm 1:** Transformer Specialization

---

**Input**  :  Fine-tuned (for the given downstream task) Transformer T, Validation set D
**Output:**  Specialized Transformer for the given downstream task T
**Function** $analyze\_element$ (*element E*) **:**
  $T_{pruned} = \text{T} - \text{E}$
  $New\_Loss = Evaluate(T_{pruned}, D)$
  **if** $New\_Loss < Min\_Loss \ and \ num\_samples\_helped > N$ **then**
    $Min\_Loss = New\_Loss$
    $\text{T} = T_{pruned}$

**Function** $num\_samples\_helped$: computes the number of samples in the validation set that
  benefit from the pruning of an element; must be $> N$ for an element to be pruned
$Baseline\_Loss = \text{Evaluate (T,D)}$
$Min\_Loss = Baseline\_Loss$
Q $= Order\_elements\_for\_inspection(T, D)$
**for** *each layer L in T* **do**
  Replace soft self-attention in L with hard self-attention
  $New\_Loss = Evaluate(T, D)$
  **if** $New\_Loss < Min\_Loss \ and \ num\_samples\_helped > N$ **then**
    $Min\_Loss = New\_Loss$
  **else**
    Restore soft self-attention in L

**while** *Q is not empty* **do**
  TrialElement = Q.pop()
  $analyze\_element$(TrialElement)
  **if** *TrialElement has not been pruned from T and* $New\_Loss < Baseline\_Loss$ **then**
    **if** *TrialElement is an attention block* **then**
      **for** *each attention head h in TrialElement* **do**
        $analyze\_element$(h)

    **else if** *TrialElement is a feed-forward block* **then**
      **for** *each neuron w in TrialElement* **do**
        $analyze\_element$(w)

return T

---

seen so far but less than the baseline loss, we inspect the element at a finer granularity, and prune only parts of the element that hinder performance (rather than pruning the entire element).

**Hierarchical processing of elements.** It is computationally prohibitive to analyze every single parameter in large Transformers using the method described in Alg 1. Since the framework iterates through the entries of the queue sequentially, its efficacy is dependent on both the total number of elements under consideration, and the time required to analyze each element. We take advantage of the inherently hierarchical structure of Transformers and consider the elements in a hierarchical manner, ordered by increasing granularity. Specifically, we analyze entire feed-forward and self-attention blocks first, and inspect them at finer granularity (attention heads and neurons) only when required. Through this ordering, we are able to quickly eliminate large numbers of parameters from further consideration. In addition, due to the over-parameterized nature of Transformers, it is likely that time-consuming blocks are pruned from the Transformer earlier in the process, thereby speeding up future iterations of the framework. For example, eliminating a single feed-forward block in the BERT-Base model removes 5.6% of all parameters under consideration, and speeds up future iterations by $1.15\times$. To further reduce the number of elements under consideration, we also dynamically remove elements if they are encompassed by a high-importance block. For example, if a given self-attention block is determined to be of high importance (the validation loss with the block removed is greater than the baseline loss), we remove all heads within that block from further consideration.

**Creating an ordered queue of elements for inspection.** Since our framework performs greedy pruning of highly over-parameterized models, it is essential to know where the harmful elements

are likely to be. Some elements may appear to be harmful for the downstream task (especially in early iterations of the framework), but this often ends up being an artifact of over-parameterization. As a result, when elements are not carefully ordered for inspection, our framework lands in local minima of the validation loss function, leading to inefficient models with sub-optimal generalization ability. Our solution to this problem utilizes the unique linguistic properties captured by the different Transformer layers (Jawahar et al., 2019) to guide the ordering of elements for inspection. For example, it was found that BERT captures phrase-level information in the lower layers, mapping related tokens together. The lower layers also capture surface features, while the middle layers capture syntactic features and higher layers capture semantic features. It was also observed that BERT requires deeper layers only when long-range dependency information is required. Different tasks require different types of linguistic knowledge. For example, sentiment analysis requires only local context, and long-range information often ends up confusing the model, since sentiments often change rapidly; it is also unlikely that syntactic and semantic information are needed. Hence, we place the final layer at the front of the queue, and work our way backwards towards the first layer, since blocks in the final layers are more likely to hinder performance on sentiment analysis. This ordering of elements ensures that elements that are pruned early in our framework (when the model is most over-parameterized) do not lead the system into bad local minima.

## 3.2 SELECTIVE USE OF HARD SELF-ATTENTION

In traditional Transformer architectures, the self-attention operation computes the attention scores of each token in the input sequence with all the other tokens. These attention scores, after passing through the softmax operation, are used to build the new representation for the token based on the extent of attention it places on the other tokens. The use of this "soft" attention enables end-to-end training of the Transformer. However, we observe that replacing the softmax operation on all attention scores with a softmax only on the attention scores above a certain threshold $T_a$ in selected layers after training and fine-tuning is completed helps the Transformer focus only on the relevant parts of the input sequence (Fig. 1). This leads to better information flow inside the model, improving performance on many downstream tasks (especially those involving sequence classification and question answering). This also introduces a large amount of activation sparsity that can be exploited to analyze the information faster, and helps alleviate the critical memory bottleneck in Transformers. $T_a$ is set to a fraction of the maximum attention score of each token, and tokens on which the selected token places significantly less attention than it places on its most relevant token are considered irrelevant. The exact value is tuned for the different downstream tasks (based on maximizing the validation loss), since we observe that different tasks require different attention spans for optimal performance. We note that replacing soft attention with hard attention in all layers (especially the deeper layers) leads to loss of important information in the model and hence, loss of accuracy, necessitating its selective use. In order to identify the layers that benefit from hard attention, we replace the soft attention with hard attention in each layer (one by one), and inspect which type of attention leads to smaller validation loss.

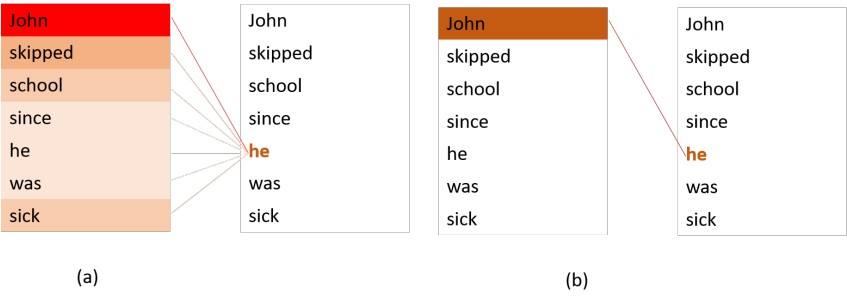

Figure 1: **Illustrations of (a) Traditional "soft" self-attention and (b) "Hard" self-attention for the word "he".** In hard self-attention, "he" concentrates all of its focus on "John", while in soft self-attention, "he" places a small amount of attention on the other irrelevant words also. Therefore, hard self-attention helps build a better representation for "he".

## 4 EXPERIMENTS AND RESULTS

We implement our techniques within Huggingface's Transformers library in PyTorch (Wolf et al., 2019). We use Intel AI's NLP Architect for experiments on Q8BERT. The experiments were performed on a GeForce RTX 2080 Ti GPU with 11GB memory. All results are reported on the test set, averaged across 10 runs with random seeds after 3 epochs of fine-tuning, unless otherwise specified. **We randomly sample** 15% **of the training set with class balance, and use it as the validation set to guide the Specialization process**. We present results on the GLUE dev set in Appendix A.

**Specialization leads to more accurate models that are also faster and smaller.** We present results on GLUE (Wang et al., 2019), a set of Language Understanding tasks, and SQUADv1.1 (Rajpurkar et al., 2016), a Question Answering task, in Table 1. For GLUE, we present results on the test set using the GLUE evaluation server to obtain the scores (Xu et al., 2020), and for SQUAD, we present results on the dev set. Specialized models are *up to 4.5% more accurate, while also being up to 2.5x faster and up to 3.2x smaller than their baseline counterparts*. In addition, Specialization is able to obtain substantial improvements over Q8BERT-base (already 4x smaller than BERT-base due to the use of 8-bit integer quantization) and DistilBERT-base (60% faster and smaller than BERT-base). Specialized Q8BERT-base and DistilBERT-base models exceed the accuracy of non-Specialized BERT-base models, while being up to 3.7x faster and 12.1x smaller than BERT-base. Therefore, Specialization can be used in conjunction with other approaches that improve the efficiency of Transformers to create highly efficient accurate models for different downstream tasks.

Table 1: **Results on GLUE and SQUAD v1.1.** We report Matthews correlation for CoLA, Pearson Correlation for STS-B and accuracy for all other tasks. We report only "matched" accuracy for MNLI and the Exact Match score for SQUAD. Speedup and Compression are reported over the non-Specialized baselines for DistilBERT and Q8BERT, and not over BERT-base.

| | SQUAD v1.1 | CoLA | MNLI | MRPC | QNLI | QQP | RTE | SST-2 | STS-B | WNLI | Average |
|---|---|---|---|---|---|---|---|---|---|---|---|
| BERT-Base | 81.97 | 51.38 | 85.37 | 83.11 | 90.74 | 89.76 | 62.65 | 95.94 | 83.8 | 64.1 | 78.88 |
| Specialized BERT-Base | **83.88** | **54.43** | **86.63** | **85.88** | **91.17** | **91.12** | **65.81** | **96.7** | **85.08** | **65.1** | **80.58** |
| Speedup/ Compression | 1.38x / 1.52x | 1.98x / 2.88x | 1.21x / 1.6x | 1.61x / 2.02x | 1.16x / 1.31x | 1.49x / 1.48x | 2.2x / 2.69x | 1.32x / 1.44x | 1.71x / 1.38x | 2.51x / 3.18x | 1.67x / 1.95x |
| Q8BERT-Base | 81.55 | 50.5 | 84.73 | 82.05 | 90.19 | 89.65 | 61.26 | 94.32 | 83.09 | 61.21 | 77.86 |
| Specialized Q8BERT-Base | **82.52** | **54.12** | **85.95** | **84.77** | **91.08** | **91.44** | **65.8** | **96.02** | **84.14** | **63.1** | **79.73** |
| Speedup/ Compression | 1.31x / 1.41x | 2.28x / 3.02x | 1.08x / 1.24x | 1.57x / 1.47x | 1.3x / 1.2x | 1.26x / 1.38x | 2.28x / 2.74x | 1.15x / 1.21x | 1.58x / 1.46x | 2.44x / 2.82x | 1.63x / 1.8x |
| DistilBERT-Base | 79.94 | 50.5 | 83.04 | 81.07 | 89.72 | 88.92 | 62.01 | 93.88 | 83.49 | 61.3 | 77.39 |
| Specialized DistilBERT-Base | **81.91** | **54.08** | **85.47** | **85.02** | **91.01** | **89.2** | **64.5** | **95.48** | **84.16** | **63.3** | **79.41** |
| Speedup/ Compression | 1.21x / 1.28x | 1.84x / 1.96x | 1.09x / 1.14x | 1.34x / 1.41x | 1.07x / 1.08x | 1.21x / 1.14x | 1.94x / 1.7x | 1.08x / 1.17x | 1.31x / 1.28x | 2.31x / 1.98x | 1.44x / 1.41x |
| XLNet-Base | 82.28 | 51.98 | 85.93 | 83.68 | 91.01 | 89.83 | 62.01 | 95.88 | 84.12 | 64.8 | 79.15 |
| Specialized XLNet-Base | **83.93** | **54.28** | **86.8** | **85.92** | **91.78** | **91.12** | **65.78** | **96.98** | **85.46** | **65.1** | **80.71** |
| Speedup/ Compression | 1.48x / 1.55x | 2.06x / 2.84x | 1.38x / 1.72x | 1.98x / 2.12x | 1.47x / 1.56x | 1.78x / 1.6x | 2.11x / 2.48x | 1.59x / 1.71x | 1.93x / 1.65x | 2.46x / 2.98x | 1.83x / 2.02x |

**Specialization reduces sensitivity to random seed initialization.** Previous research (Dodge et al., 2020) has shown that the random seed (which determines the initialization of the task-specific layer and the order of training data for the downstream task) has a significant impact on the quality of the fine-tuned models. When the amount of training data is small (which is the case with most downstream NLP tasks), this effect becomes more pronounced, evidenced by the fact that WNLI, which has the least amount of training samples (634 training samples), exhibits highest variance across runs (Table 2). Therefore, in order to find a model with high accuracy on a downstream task, multiple models need to be created (using different random seeds), and evaluated. Here, we demonstrate that Specialized models exhibit significantly less sensitivity to random seeds than their non-Specialized counterparts (Table 2), thereby greatly increasing the odds of finding "good" models in fewer iterations. In particular, we find that even if two models have vastly different accuracies

on the validation set before Specialization, they converge to similar accuracies after Specialization. This, in turn, reduces variance on the test set.

Table 2: **Sensitivity to random seeds.** Results reported are averaged across the GLUE tasks and SQUAD on the Base models of each Transformer.

| Transformer | Average variance across tasks | Maximum variance for a single task | Average of maximum task scores | Average of minimum task scores |
|---|---|---|---|---|
| BERT | 1.48 | 4.86 (WNLI) | 80.05 | 76.53 |
| Specialized BERT | 0.86 | 1.95 (WNLI) | 82.02 | 79.93 |
| Q8BERT | 1.61 | 4.62 (WNLI) | 79.01 | 75.29 |
| Specialized Q8BERT | 1.1 | 2.02 (WNLI) | 81.14 | 79.58 |
| DistilBERT | 1.29 | 3.85 (WNLI) | 78.78 | 74.89 |
| Specialized DistilBERT | 0.83 | 1.78 (WNLI) | 80.96 | 79.59 |

**Larger models Specialize better.** Current state-of-the-art Transformer networks, such as T5 (Raffel et al., 2019) and GPT-3 (Brown et al., 2020), have hundreds of billions of parameters. Model sizes are also expected to grow further in the future as increasing the number of parameters has been shown to improve performance. This makes it computationally challenging to train Transformers as well as perform inference using them. Recent research (Li et al., 2020) has shown that larger models converge in a significantly smaller number of training iterations than smaller models, and hence they train faster in spite of requiring more time per iteration. However, larger models are significantly slower than smaller models at inference time. Here, we demonstrate that Specialized larger models achieve much higher accuracy while being comparable in speed to Specialized smaller models (Table 3). In addition to having a higher accuracy ceiling, we also demonstrate that Specialized BERT-Large (pruned beyond the max accuracy point, by relaxing the constraints to trade-off accuracy for efficiency), is 1.5x faster than Specialized BERT-base at iso-accuracy. This indicates that larger models capture greater linguistic knowledge, but also contain more redundant parameters for the different downstream tasks, further motivating the development of larger and better language models that can be Specialized for fast and accurate inference.

Table 3: **Specialization of BERT-base and BERT-large.** Results reported are averaged across the GLUE tasks and SQUAD.

| Transformer | Average accuracy | Average samples/second |
|---|---|---|
| BERT-Base | 78.88 | 0.21 |
| Specialized BERT-Base | 80.58 | 0.12 |
| BERT-Large | 80.98 | 0.45 |
| Specialized BERT-Large | 82.89 | 0.15 |

**Previously proposed techniques to improve inference efficiency are more effective when applied to Specialized models.** Specialization identifies and prunes harmful elements of the Transformer for the downstream task at hand. Techniques that prune elements that have minimal impact on the output are complementary to our techniques, and we demonstrate this using the popular Lottery Ticket Hypothesis (Frankle & Carbin, 2019), which finds sparse sub-networks that can be trained to match the performance of the large network. Lottery Ticket Hypothesis has been successfully applied to Transformers also (Prasanna et al., 2020; Liang et al., 2021; Chen et al., 2020), and using these techniques, we demonstrate that winning tickets of the Specialized model are consistently more accurate than the iso-efficient winning tickets of the conventional fine-tuned models (Figure 2).

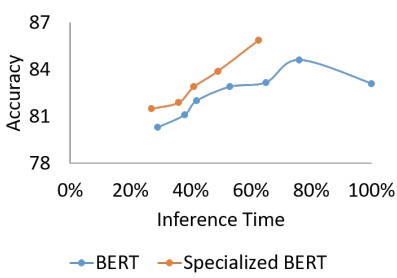

Figure 2: **The winning tickets of conventional fine-tuned and Specialized BERT-Base on MRPC.**

**Specialization provides insights into the working of Transformers.** We analyze which elements of the Transformer are pruned for different downstream tasks using different Transformer models (Fig. 3). We find that the differences in importance of elements are more pronounced across different tasks than across different models. For example, for sentiment analysis, long-range dependency information is not required, and often ends up confusing the model. Hence, for all models fine-tuned for sentiment analysis, we observe that components in later layers (closer to the output) are more likely to be pruned. This is not the case with Language Modelling tasks (predicting masked words in text), where longer attention spans are required. Across models, we only observe subtle differences. For example, we find that XLNet (auto-regressive) is able to learn important task-specific information earlier than BERT (non auto-regressive), similar to the observation made in (Sajjad et al., 2020). Hence, we are able to drop more components (in earlier layers) in XLNet than in BERT, leading to more efficient models for inference. In DistilBERT (a distilled model), we find that there is a clear demarcation in linguistic knowledge across layers due to the reduced capacity of the model. This is evidenced by the fact that elements in the top four layers are never pruned across all Language Understanding tasks, while the boundaries are more soft in the original models. We also observe that Hard Attention is most useful in the lower layers, where phrase-level information is captured, for all the models and tasks. As future work, our framework can be combined with previously proposed techniques (such as probing classifiers) to gain deeper understanding of the working of Transformers, especially at finer levels of granularity.

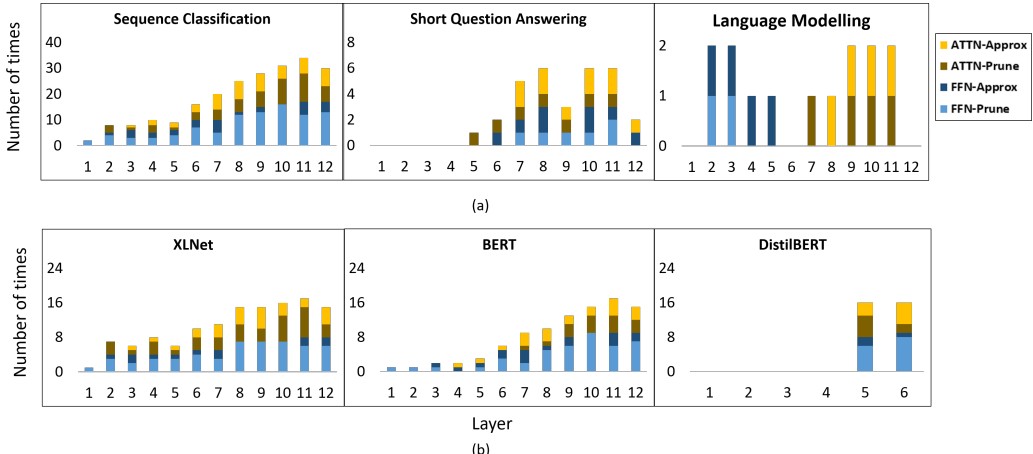

Figure 3: **Elements pruned for (a) downstream tasks and (b) Transformer architectures across GLUE and SQUAD in our most accurate models for each task.** Here, ATTN-Approx means only certain attention heads inside the attention block are pruned, and FFN-Approx means only certain neurons inside the feed-forward block are pruned.

**Specialization using previously proposed importance estimation techniques leads to less accurate models.** L1-norm and magnitude based pruning techniques are designed to prune elements that have minimal impact on the output, and hence, they are not effective at identifying and pruning elements that have most detrimental impact on output. Gradient-based methods such as Taylor expansion are not reliable for increasing accuracy on the downstream tasks, since Transformers cannot be repeatedly fine-tuned to recover the accuracy losses from approximating the model (they very quickly overfit the limited training data for the downstream tasks). In addition, while knowledge distillation is complementary to our method (as evidenced by our results on DistilBERT), we show that the accuracy gains from Specialization are significantly higher than those from regularized knowledge distillation proposed in (Hou et al., 2020). Super-tickets (Liang et al., 2021) demonstrates that accuracy can be improved by finding winning lottery tickets at low levels of pruning. However, we find that alleviating the over-parameterization issue by pruning elements that hinder performance (and the selective use of hard attention) leads to more accurate models than by pruning elements that have least impact on output in the original model (Table 4).

**Specialization incurs minimal overhead.** Our Specialization framework does not require any additional training or fine-tuning iterations. It only requires multiple passes through a small validation

set. Our hierarchical processing ensures that the number of elements to inspect is greatly reduced. In addition, each pass is expected to become progressively faster, since the framework potentially prunes an element in each iteration, and our ordering of elements ensures that large, time-consuming blocks are pruned early in the process, making future iterations faster. Therefore, the overheads of Specialization are negligible compared to the time required for re-training or additional fine-tuning. We find that the average (for GLUE and SQUAD) wall-clock time for Specialization is only 7.6 minutes for BERT-base and 18.3 minutes for BERT-large on a single GPU.

We also evaluate the heuristics used in our framework (Table 4). When elements are not carefully ordered for inspection, we find that elements that are pruned early in the Specialization process are pruned due to over-parameterization, and not because the linguistic knowledge contained in these elements is harmful/not useful for the task at hand. This leads the system into a bad local minima, where the validation loss cannot be further reduced by pruning other elements. We observe that starting at the layer granularity leads to worse models than starting at the block granularity. This is because the attention and feed-forward blocks in each layer have vastly differently functionality, and hence, the effect of removing an attention block of relatively high significance is often masked by removing the feed-forward block that greatly hinders performance in the same layer (and vice-versa), causing the entire layer to be pruned. This destabilizes the Specialization process, and also leads the system into bad local minima, thereby creating inferior models. To drive home the fact that our greedy approach combined with a global error bound does not lead to inferior models, we also experiment with an adaptive loss threshold. In particular, we use a very tight constraint when analyzing elements at coarse granularities, and relax the constraint as we move towards finer granularities. We again find that there is negligible change in the quality of the final model produced (the accuracy is slightly lower, possibly due to over-fitting to the validation set), but the Specialization process is significantly slower. We hypothesize that a single global error bound is sufficient because we order the elements in such a way that for the given task at hand, we intuitively expect that the elements at the head of the queue are likely to be removed using the linguistic knowledge in different layers.

Table 4: **[Left] Results of Specialization with previously proposed methods (MRPC with BERT-Base).** For L1-norm, magnitude and Taylor (absolute gradient of loss), we prune 5% of the least important weights at a time and record the test accuracy. We report the highest test accuracy seen in this process. **[Right] Results of Specialization with different heuristics.** All heuristics use hierarchical processing of ordered elements with Selective Hard Attention, unless otherwise specified.

| Pruning Method | Accuracy |
|---|---|
| Baseline (No Pruning) | 83.11 |
| L1-Norm | 83.16 |
| Magnitude | 83.18 |
| Taylor Expansion (with additional fine-tuning) | 83.19 |
| Taylor Expansion (no additional fine-tuning) | 83.32 |
| Regularized Knowledge Distillation (DynaBERT) | 83.46 |
| Super-ticket | 84.62 |
| **Ours** | **85.88** |

| Heuristic | Specialized Model Accuracy | Wall-clock Specialization Time (minutes) |
|---|---|---|
| Randomly ordered elements | 83.42 | 7.2 |
| Start by inspecting layers | 83.44 | 5.8 |
| No Selective Hard Attention | 84.94 | 6.3 |
| Adaptive Threshold | 85.82 | 29.3 |
| **Ours** | **85.88** | **6.6** |

## 5 CONCLUSION

We proposed a Specialization framework to optimize fine-tuned Transformers for the different downstream tasks. The framework identifies elements that are hinder performance on the downstream task at hand, and prunes these elements. We also demonstrated the advantage of selectively using hard self-attention in selected layers to improve information flow. Using this framework, we produced models that were guaranteeably more accurate, while also being up to faster and smaller.

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

# A   ADDITIONAL EXPERIMENTS AND STUDIES

## A.1   RESULTS ON THE GLUE DEVELOPMENT SET.

We present additional results on the GLUE dev set using the same models used to present results on the test set in Table 5. We use 15% of the training set as the validation set for Specialization. Similar to results on the test set, Specialized models are consistently more accurate, faster and smaller than their conventional fine-tuned counterparts. In addition, Specialization is able to obtain substantial improvements over already optimized models (Q8BERT and DistilBERT).

Table 5: **Results on GLUE.** We report Matthews correlation for CoLA, Pearson Correlation for STS-B and accuracy for all other tasks. We report only "matched" accuracy for MNLI. Speedup and Compression are reported over the non-Specialized baselines for DistilBERT and Q8BERT, and not over BERT-base.

|  | CoLA | MNLI | MRPC | QNLI | QQP | RTE | SST-2 | STS-B | WNLI | Average |
|---|---|---|---|---|---|---|---|---|---|---|
| BERT-Base | 53.21 | 84.43 | 86.11 | 90.68 | 91.06 | 64.25 | 93.23 | 88.13 | 56.11 | 78.58 |
| Specialized BERT-Base | **55.45** | **85.24** | **88.89** | **91.63** | **91.77** | **64.8** | **94.56** | **89.41** | **58.82** | **80.06** |
| Speedup/ Compression | 1.98x / 2.88x | 1.21x / 1.6x | 1.61x / 2.02x | 1.16x / 1.31x | 1.49x / 1.48x | 2.2x / 2.69x | 1.32x / 1.44x | 1.71x / 1.38x | 2.51x / 3.18x | 1.67x / 1.95x |
| Q8BERT-Base | 52.85 | 83.73 | 85.05 | 91.19 | 90.65 | 64.26 | 92.32 | 88.09 | 56.11 | 78.25 |
| Specialized Q8BERT-Base | **55.45** | **84.32** | **86.79** | **91.96** | **91.57** | **64.8** | **93.16** | **88.97** | **57.12** | **79.35** |
| Speedup/ Compression | 2.28x / 3.02x | 1.08x / 1.24x | 1.57x / 1.47x | 1.3x / 1.2x | 1.26x / 1.38x | 2.28x / 2.74x | 1.15x / 1.21x | 1.58x / 1.46x | 2.44x / 2.82x | 1.63x / 1.8x |
| DistilBERT-Base | 50.25 | 82.04 | 84.07 | 88.72 | 89.92 | 61.01 | 90.48 | 86.49 | 56.33 | 76.59 |
| Specialized DistilBERT-Base | **54.1** | **83.48** | **86.24** | **89.65** | **90.98** | **63.94** | **92.1** | **87.98** | **57.08** | **78.41** |
| Speedup/ Compression | 1.84x / 1.96x | 1.09x / 1.14x | 1.34x / 1.41x | 1.07x / 1.08x | 1.21x / 1.14x | 1.94x / 1.7x | 1.08x / 1.17x | 1.31x / 1.28x | 2.31x / 1.98x | 1.44x / 1.41x |
| XLNet-Base | 57.89 | 86.97 | 87.61 | 90.18 | 91.11 | 59.88 | 93.62 | 87.93 | 56.48 | 79.08 |
| Specialized XLNet-Base | 57.89 | **87.81** | **88.96** | **91.84** | **92.08** | **64.8** | **94.74** | **88.71** | **58.8** | **80.61** |
| Speedup/ Compression | 2.06x / 2.84x | 1.38x / 1.72x | 1.98x / 2.12x | 1.47x / 1.56x | 1.78x / 1.6x | 2.11x / 2.48x | 1.59x / 1.71x | 1.93x / 1.65x | 2.46x / 2.98x | 1.83x / 2.02x |

## A.2   SIGNIFICANCE OF ORDERING ELEMENTS FOR INSPECTION

We find that the order in which elements of the fine-tuned model are inspected during accuracy-driven pruning has a large impact on the generalization performance of the Specialized model. Prior research (Jawahar et al., 2019) has shown that the linguistic knowledge contained in different Transformer layers can be demarcated into three main functional regions: phrase-level information in bottom layers (the first one-third of transformer layers closest to the input), semantic and syntactic information in the middle layers, and long-range dependency information in the top layers. When elements are not carefully ordered for inspection, we find that elements that are pruned early in the Specialization process are pruned due to over-parameterization, and not because the linguistic knowledge contained in these elements is harmful/not useful for the task at hand. This leads the system into a bad local minima, where the validation loss cannot be further reduced by pruning other elements (Table 6). The quality of the Specialized model is worst when the blocks containing the most important linguistic knowledge is inspected first (bottom layers for sentiment analysis, see Appendix A.3), since these blocks may get pruned simply to alleviate the over-parameterization. In addition, we find that the ordering of elements within the same functional block (for example, blocks in layer 10 before layer 11 or vice-versa), has negligible impact on the quality of the Specialized model, further demonstrating the existence of regions with different kinds of linguistic knowledge.

Table 6: **Specialized accuracy from inspecting elements in different orders on SST-2 using BERT-Base.** The baseline validation and dev accuracy of the fine-tuned model are 93.98 and 93.23, respectively.

| Order in which elements are inspected | Validation accuracy of Specialized model | Dev accuracy of Specialized model |
|---|---|---|
| Bottom, Middle, Top | 94.06 | 93.28 |
| Bottom, Top, Middle | 94.38 | 93.41 |
| Middle, Bottom, Top | 94.44 | 93.52 |
| Middle, Top, Bottom | 95.22 | 94.28 |
| Top, Bottom, Middle | 95.18 | 93.88 |
| **Top, Middle, Bottom** | **95.24** | **94.56** |

## A.3 ANALYZING THE LINGUISTIC KNOWLEDGE REQUIRED FOR DIFFERENT DOWNSTREAM TASKS

Different downstream tasks require different types of linguistic knowledge. We find that the most important functional block of the Transformer model for each task (corresponding to the most important kind of linguistic knowledge required to effectively solve the task) agrees with our intuition about the task. For example, we expect sentiment analysis to require only local context, since long-range information often ends up confusing the model (sentiments often change rapidly); it is also unlikely that syntactic and semantic information are needed. This is experimentally validated by the fact that inspecting the blocks in top layers, followed by the middle and finally the bottom layers leads to most accurate models on SST-2 (Table 7). Similarly, we expect knowledge about language syntax and semantics to be most important for a task that tests the linguistic acceptability of a given sentence (CoLA). We also find that this holds across all Transformer architectures that we study in this work (BERT, DistilBERT, Q8BERT and XLNet). We leverage this intuition about the different NLP tasks in our framework to reduce the overheads of Specialization without compromising on the quality of the Specialized model. If there is no prior knowledge of the task, we need to inspect elements in multiple orders to obtain the best model.

Table 7: **The order of inspected elements that provides maximum accuracy for different downstream tasks using BERT-Base, DistilBERT-Base, Q8BERT-Base and XLNet-Base.** We find that the same ordering provides best performance on all the studied Transformer architectures.

| Dataset | Task | Best ordering | Most important type of linguistic knowledge for solving this task |
|---|---|---|---|
| CoLA | Linguistic acceptability | Top, Bottom, Middle | Semantic/ Syntactic |
| MNLI | Entailment | Top, Middle, Bottom | Phrase-level |
| MRPC | Semantic equivalence | Top, Middle, Bottom | Phrase-level |
| QNLI | Question answering | Top, Middle, Bottom | Phrase-level |
| QQP | Semantic equivalence | Top, Middle, Bottom | Phrase-level |
| RTE | Entailment | Top, Middle, Bottom | Phrase-level |
| SST-2 | Sentiment Analysis | Top, Middle, Bottom | Phrase-level |
| STS-B | Sentence similarity | Top, Middle, Bottom | Phrase-level |
| WNLI | Reading comprehension | Top, Middle, Bottom | Phrase-level |
| SQUAD | Question answering | Top, Middle, Bottom | Phrase-level |

### A.4    ANALYZING THE BENEFITS OF SPECIALIZATION

Transformers are pre-trained on a large text corpus on a difficult NLP task, such as predicting the next word in text given all the preceding words. When pre-trained models are fine-tuned for different downstream tasks, they contain lots of task-irrelevant information that add noise and confuse the model. In addition, since the pre-trained models are highly overparameterized, they severely overfit the small fine-tuning datasets. Intuitively, Specialization helps the model focus on the task at hand by eliminating the irrelevant information in the fine-tuned models.

#### A.4.1    IMPROVED GENERALIZATION

Accuracy-driven pruning can be seen as a form of training, since the objective of accuracy-driven pruning – minimizing loss on the (validation) dataset – is exactly the same as the objective of training. When large, over-parameterized models are trained on very limited data, the constraints of accuracy-driven pruning – (1) loss can be reduced only be setting weights to 0 (equivalent to pruning elements), and no other changes to weights are allowed, and (2) a majority of samples in the validation set must benefit from each weight update; reduction in loss is a necessary but insufficient condition for weights to get updated – help it learn more effectively than SGD. As a result, we find that training with SGD on a subset of the training data, followed by accuracy-driven pruning on the remaining unseen data, produces models with better generalization performance than training with SGD on the entire dataset (Figure 4). As an added benefit, accuracy-driven pruning also leads to smaller and faster models at inference time.

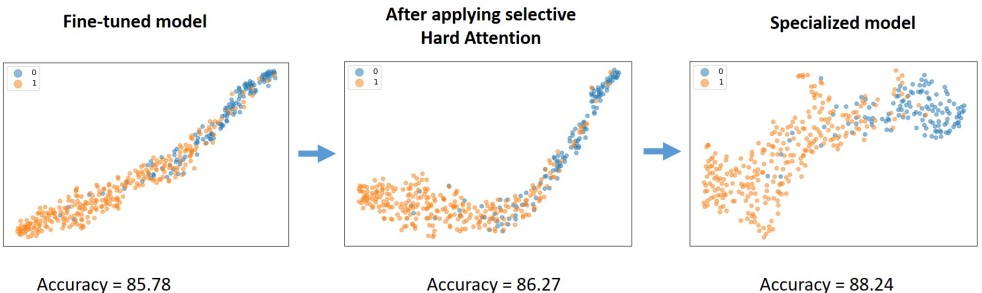

Figure 4: **Change in class boundaries during Specialization on MRPC with BERT-Base.** We use T-distributed Stochastic Neighbor Embedding (TSNE) on word embeddings right before the final classifier, using the 3 main components of each embedding. Here, 0 and 1 refer to sentence pairs that are semantically non-equivalent and equivalent, respectively. Specialized models show better class separability, and hence, are more accurate.

#### A.4.2    REDUCED VARIANCE

Accuracy-driven pruning and selective Hard Attention filter out noise in the model by removing irrelevant information that ends up confusing the model. Accuracy-driven pruning reduces noise by removing task-irrelevant information. Selective Hard Attention further reduces noise by ensuring that the model focuses only on the relevant parts of the input. We find that this leads to reduced variance on the dev/test sets, where Specialized models from different seeds acquire similar word representations, even if the initial fine-tuned models have vastly different word representations (and hence accuracies). (Figure 5). However, the reduction in variance is limited by the effects of random initialization of the task-specific final layer.

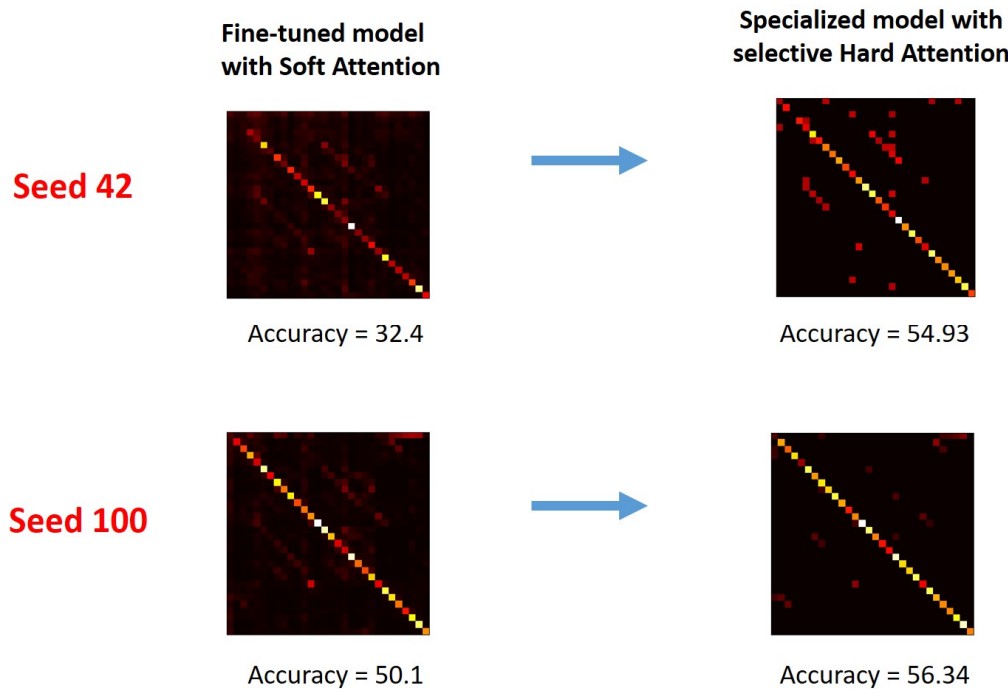

Figure 5: **Attention patterns of Specialized models on WNLI using BERT-Base, shown at the final ATTN block in the model (layer 12 in the fine-tuned model, layer 9 in the Specialized model).** WNLI has the smallest training set among all of our studied tasks, and hence, exhibits highest sensitivity to random seeds.

