# OpenReview forum: "Specialized Transformers: Faster, Smaller and more Accurate NLP Models"
_ICLR.cc/2022/Conference — ICLR 2022 Submitted_

### Official Review · Reviewer_bSmL · 2021-11-02

**Correctness:** 4
**Technical Novelty And Significance:** 3
**Empirical Novelty And Significance:** 3
**Recommendation:** 8
**Confidence:** 4

**Main Review:**

### Pros
1. The paper is overall well-written and easy to follow.
2. The evaluation is thorough with SQUAD and GLUE. The method shows effectiveness on both benchmarks and with different models by outperforming the fine-tuned counterparts.
3. The paper proposes a progressive and hierarchical strategy that provides a solution for both self-attention and FFNN in a Transformer, which has the potential to be adopted in more tasks (e.g., generation, translation).
4. Figure 3 is quite interesting - it looks like DistilBERT is more "dense" than an uncompressed model and it warrants further exploration.
5. The method is practical since it does not require re-training and works on downstream tasks thus enabling the reuse of nearly all off-the-shelf pretrained models.

### Cons
1. My main concern is novelty - the pruning technique is not new and pruning a neural network progressively is also not a novel idea. That being said, the paper is still informative in my opinion.
2. Although specialization does not require a complete re-training, it does introduce extra overheads for training such a model. The authors mentioned this problem on page 8 but I'd like to see more discussion with analysis.

### Missing References
- https://arxiv.org/abs/2002.10957
- https://arxiv.org/abs/2005.07683
- https://arxiv.org/abs/2006.04152
- https://arxiv.org/abs/2109.04838

**Summary Of The Paper:**

This paper proposes a new framework for pruning a pretrained Transformer on downstream tasks for better performance and efficiency. The resulted model is 2.5x faster and 3.2x smaller than a fine-tuned model. The authors test the effectiveness on GLUE and SQUAD with BERT-base, Q8BERT, DistilBERT and XLNet.

**Summary Of The Review:**

This paper proposes a new pruning framework for pretrained Transformers with good evaluation and analysis. Although the idea is not completely new, I would like to recommend weak acceptance for this paper.

---

> ### Author Response · Authors · 2021-11-15
> **Response to Reviewer bSmL**
>
> We thank the reviewer for their thoughtful review. We address the main concerns below.
>
> **Re: My main concern is novelty - the pruning technique is not new and pruning a neural network progressively is also not a novel idea.**
>
> We agree that greedy pruning methods and progressive model pruning have been previously proposed. However, we would like to underscore that the novelty of our approach lies in the pruning objective. Previous pruning methods prune elements that have minimal impact on the model output (in order to improve efficiency with minimal loss in accuracy). This approach fundamentally cannot achieve accuracy improvements. Our accuracy-driven pruning method instead prunes elements that have a detrimental effect on the output (larger the better). This, in turn, leads to models that are more accurate while also being smaller and faster, as opposed to the accuracy-efficiency trade-offs typically involved in other pruning methods. In fact, contrary to other pruning methods, accuracy-driven pruning can be seen as a form of training, since the objective of accuracy-driven pruning -- minimizing loss on the (validation) dataset -- is exactly the same as the objective of training. When large, over-parameterized models are trained on very limited data, the constraints of accuracy-driven pruning -- (1) loss can be reduced only be setting weights to 0 (equivalent to pruning elements), and no other changes to weights are allowed, and (2) a majority of samples in the validation set must benefit from each weight update; reduction in loss is a necessary but insufficient condition for weights to get updated -- help it learn more effectively than SGD. As a result, we find that training with SGD on a subset of the training data, followed by accuracy-driven pruning on the remaining unseen data, produces models with better generalization performance than training with SGD on the entire dataset.
>
> In addition, previously proposed techniques are also complementary to our method, and we demonstrate this in the sub-section of our results section titled “Previously proposed techniques to improve inference efficiency are more effective when applied to Specialized models”. Here, we show (see Figure 2 on page 7) that conventional pruning can be applied after Specialization to produce models that are even more efficient for a given accuracy constraint.
>
> **Re: Although specialization does not require a complete re-training, it does introduce extra overheads for training such a model. The authors mentioned this problem on page 8 but I'd like to see more discussion with analysis.**
>
> Yes, our Specialization framework adds post fine-tuning overheads. We provide an analysis of how the different design choices in our Specialization framework (hierarchical processing, ordering of elements for inspection) contribute towards reducing the overheads. We also provide the average Specialization time required for these tasks in the updated paper (see “Specialization incurs minimal overhead” heading within the results section on page 9). If the reviewer would like to see any other analyses, we will be happy to perform the required experiments/ studies.
>
> We have also added the references, and thank the reviewer for pointing them out. We hope we have addressed the main concerns. If not, please let us know and we will be happy to address them in subsequent responses and revisions.

---

> > ### Comment · Reviewer_bSmL · 2021-11-21
> > **Recommendation update**
> >
> > Thanks for the response!
> >
> > The authors' response answers my questions and I've updated my recommendation from 6 (marginally above threshold) to 8 (good paper, accept).

---

### Official Review · Reviewer_GDsL · 2021-11-02

**Correctness:** 2
**Technical Novelty And Significance:** 2
**Empirical Novelty And Significance:** 3
**Recommendation:** 5
**Confidence:** 3

**Main Review:**

The method is a *greedy* method with unjustified heuristics. However, how can it perform well without getting stuck in a local minimum and have low variance? I am pretty suspicious about the results because many essential details and explanations are missing.

As far as I understand, it prunes a transformer fine-tuned for a downstream task. (I’m a little bit confusing because the abstract mentions pruning a pre-trained transformer.) Then, it is already overfitted in some sense. I am not sure how pruning can improve the generalization.

I am curious whether similar pruned architectures are obtained with different random seeds. Is the pruning process deterministic or stochastic? If it is deterministic, is it measuring with differently fine-tuned transformers? Do you have any explanation about low variance?

Are pruning and hard self-attention separate each other? What is the order of them?

The method does not perform any retraining. Simply thinking, eliminating one layer or even a smaller portion of parameters could change the representation distribution of the following layers. How can it perform well?

The paper should be revised further. The related work section is just a listing of works that improve the efficiency of the transformer. The authors should categorize them and provide clear relation with their work. Method and result sections include many long paragraphs only divided by headings.

I don’t understand how to use Taylor expansion to estimate the importance of each parameter. Is it more than a first-order gradient?

**Summary Of The Paper:**

This paper proposes Specialized Transformer that identifies harmful parts and prunes a pre-trained transformer in a greedy and hierarchical manner to boost accuracy in a downstream task. According to the paper, the proposed Specialized Transformer is faster, smaller, and more accurate than a standard fine-tuned model without any retraining.

**Summary Of The Review:**

The results look very promising, but the presentation is bad. The method lacks crucial details for implementation. Their methods seem too naive to be effective, but the authors do not fully explain why they result in a good performance.

---

> ### Author Response · Authors · 2021-11-15
> **Response to Reviewer GDsL (Part 1)**
>
> We thank the reviewer for their thoughtful review. We address their main concerns below.
>
>
> **Re: As far as I understand, it prunes a transformer fine-tuned for a downstream task.**
>
> Yes, this is correct. We thank the reviewer for pointing out this mistake in the abstract, and we have corrected this in the revised paper.
>
> **Re: How can pruning improve generalization performance and reduce variance?**
>
> We thank the reviewer for raising this important question. We have addressed this in Appendix A.4 in the revised version. Transformers are pre-trained on a large text corpus on a difficult NLP task. When pre-trained models are fine-tuned for different downstream tasks, they contain lots of task-irrelevant information that add noise and confuse the model. The proposed techniques (Accuracy-driven pruning and selective Hard Attention) filter out noise in the model by removing the irrelevant information, in addition to the regularization effects of Specialization. Accuracy-driven pruning reduces noise by removing task-irrelevant information. Selective Hard Attention further reduces noise by ensuring that the model focuses only on the relevant parts of the input. We find that this leads to reduced variance on the dev/test sets, where Specialized models from different seeds acquire similar word representations, even if the initial fine-tuned models have vastly different word representations (and hence accuracies). However, the reduction in variance is limited by the effects of random initialization of the task-specific final layer.
>
> Accuracy-driven pruning can also be seen as a form of training, since the objective of accuracy-driven pruning -- minimizing loss on the (validation) dataset -- is exactly the same as the objective of training. When large, over-parameterized models are trained on very limited data, the constraints of accuracy-driven pruning -- (1) loss can be reduced only be setting weights to 0 (equivalent to pruning elements), and no other changes to weights are allowed, and (2) a majority of samples in the validation set must benefit from each weight update; reduction in loss is a necessary but insufficient condition for weights to get updated -- help it learn more effectively than SGD. As a result, we find that training with SGD on a subset of the training data, followed by accuracy-driven pruning on the remaining unseen data, produces models with better generalization performance than training with SGD on the entire dataset. Overfitting is alleviated to an extent, since gain in validation accuracy often comes with a small increase in training loss due to the regularization effects of Specialization. In fact, on some datasets with a very small number of samples, training loss becomes very close to 0 after fine-tuning, while validation accuracy is very low, indicating severe overfitting. After Specialization, we see a large increase in validation accuracy with some increase in test loss, which leads to better generalization. This is a very important concern brought up by the reviewer, and we hope that this explanation and additional analyses in the appendix provide more clarity. We can also share our models and testing code if it would help the reviewer analyze the differences between the fine-tuned and Specialized models.
>
> **Re:  Eliminating one layer or even a smaller portion of parameters could change the representation distribution of the following layers. How can it perform well?**
>
> Yes, but we observe that this change is a positive change that leads to better word representations to help the model solve the task at hand. Since the goal of accuracy-driven pruning is to find a model with minimum loss on the validation set, pruning an element from the model can be viewed as a step in the right direction towards a more accurate model with better generalization ability. This is similar to how representations change for the better during each epoch of conventional training with SGD.
>
> **Re: How can the greedy method not get stuck in local minima?**
>
> There is no guarantee that accuracy-driven pruning will find the global minimum, and this is the case even with SGD or any pruning technique. Our goal is to improve the quality of the local minima that the method finds. Our experiments show that Specialization can provide substantial improvements over the baseline, indicating that it can find “better local minima”.

---

> > ### Author Response · Authors · 2021-11-15
> > **Response to Reviewer GDsL (Part 2)**
> >
> > **Re: I am curious whether similar pruned architectures are obtained with different random seeds. Is the pruning process deterministic or stochastic?**
> >
> > The pruning process is deterministic due to our ordering of elements for inspection. For a given random seed (and fine-tuned network), Specialization always produces the same final Specialized model. Across different random seeds, the final models are similar, but not exactly the same. At a coarse granularity, the same blocks are pruned (or partially pruned), but there are differences at finer granularities. For example, if the ATTN block in layer 11 is pruned with one random seed, then we find that it is pruned with other random seeds also. However, if attention head 10 in ATTN block 10 is pruned with one seed, it is not necessarily pruned with other seeds (for example, attention head 8 may be pruned with another seed). We believe that this difference is due to the differently initialized task-specific final layer and differently ordered training sets of different seeds.
> >
> > **Re: Are pruning and hard self-attention separate each other? What is the order of them?**
> >
> > Yes, accuracy-driven pruning and hard attention are separate from each other. We first selectively replace soft attention with hard attention, and then prune. This ensures that blocks that benefit from hard attention are not pruned before their attention mechanisms can be replaced.
> >
> > **Re: I don’t understand how to use Taylor expansion to estimate the importance of each parameter. Is it more than a first-order gradient?**
> >
> > We follow the approach described in “Pruning convolutional neural networks for resource efficient inference” (Molchanov et.al., 2017). The importance of a parameter h is characterized by the change in the loss function when it is removed from the model.
> > ∣∣∆C(h)∣∣ = ∣∣C(D,h = 0) −C(D,h)∣∣, which can be approximated using first-order Taylor expansion as ∣∣∆C(h)∣∣ = ∣$\frac{δC}{δh}$h∣. Then, the least important parameters are removed from the model.
> >
> > We have also reformatted the related works section. We hope we have addressed the main concerns. If not, please let us know and we will be happy to address them in subsequent responses and revisions.

---

> ### Author Response · Authors · 2021-11-27
> **Follow-up**
>
> We appreciate your insightful comments about our work. Since the discussion period is almost over, we were wondering if our rebuttal addressed your concerns. If not, please let us know and we will be happy to address them in subsequent responses.

---

> > ### Comment · Reviewer_GDsL · 2021-11-30
> > **Still confusing about why it works well**
> >
> > Thanks for the detailed answers, and sorry for the late reply!
> > The results are surprisingly promising. However, I still believe that the methods are too naive to get good accuracy.
> > It is difficult for me to fill in the gap on how the proposed method can perform remarkably well without sophisticated techniques such as re-training or advanced search.

---

> > > ### Comment · Area_Chair_qtyJ · 2021-11-30
> > > **more details?**
> > >
> > > Thanks for participating in the discussion! Could you provide any more details on what experiments or analysis the authors might be able to include to convince you of the validity of the results?

---

> > > > ### Author Response · Authors · 2021-12-01
> > > > **We are happy to provide additional clarifications**
> > > >
> > > > Thank you for the response! We would like to underscore that **accuracy-driven pruning is in fact a form of training**, since the objective of accuracy-driven pruning is to minimize loss on a specific dataset. When large, over-parameterized Transformers are trained on very limited fine-tuning data, the weight updates during SGD very quickly reduce the loss on the training data to zero due to the large capacity of the models compared to the amount of training data available. This indicates that these models memorize the training data, which leads to sub-optimal generalization performance.
> > > >
> > > > In contrast, accuracy-driven pruning is a constrained form of training. We first train on a subset of the available training data using SGD to train the task-specific final layer, and for data shaping to better match the target domain. Then, we further train the model on the unseen subset using accuracy-driven pruning. Here, weights are updated *if and only if* (a) loss on the dataset is reduced, **and** (b) the weight update leads to a model with better generalization performance (a majority of samples in the unseen dataset must benefit from each weight update). In addition, we limit the degrees of freedom of weight updates: weights can be set to 0 (equivalent to pruning elements), or retained as-is; no other changes are allowed. This prevents models from memorizing training data, and ensures that weight updates benefit a majority of samples in the unseen dataset, leading to models with better generalization ability.
> > > >
> > > > **We also note that previous works**, such as "Are sixteen heads better than one?" (https://arxiv.org/pdf/1905.10650.pdf), **have shown that performance on downstream tasks can be improved by pruning attention heads of the fine-tuned model without any retraining.** While improving accuracy on downstream tasks was not the goal of these works and the performance gains are highly inconsistent, they observe this phenomenon as a result of over-parameterization. We find that this holds at different levels of granularity, and introduce a framework that can maximize accuracy on the downstream task by pruning elements that hinder performance in an *automated and principled* manner. We also plan to release our models and testing code shortly so that the results can be empirically validated. We hope this addresses your concerns, and as mentioned by the Area Chair, we are happy to provide additional experiments and analyses in the final version, and engage in further discussions.

---

> > > > > ### Comment · Reviewer_GDsL · 2021-12-01
> > > > > **Raised my score**
> > > > >
> > > > > The authors' response partly addressed my concerns, so I raised my score.
> > > > > Looking forward to their implementation to reproduce their results.

---

### Official Review · Reviewer_Hc1r · 2021-11-03

**Correctness:** 3
**Technical Novelty And Significance:** 2
**Empirical Novelty And Significance:** 3
**Recommendation:** 3
**Confidence:** 4

**Main Review:**

Pros:
1. The proposed method does not require further fine-tuning.
2. It is simple enough, easy to apply on the general transformer-based network.
3. Speed, size, and performance could be improved at the same time. That is interesting.

Cons:
1. The pruning method is not very novel.
2. Here I quote from the paper, "where N is set to be greater than at least three-fourths the number of samples in the dataset. In order to reduce the effect of outliers, the exact value of N is tuned for the different datasets, based on the validation loss of each sample." Could you introduce the details of tuning N?
3. Lack mechanism analysis. For example, how does the attention to each word change after the pruning of each element?
4. Not enough ablation study. Not sure what really matters. The pruning of which element matters the most?
5. "For example, sentiment analysis requires only local context, and long-range information often ends up confusing the model, since sentiments often change rapidly; it is also unlikely that syntatic and symantic information are needed. Hence, we place the final layer at the front of the queue, and work our way backwards towards the first layer, since blocks in the final layers are more likely to hinder performance on sentiment analysis". Here I have two questions: 1. How do you get this conclusion about the final layers and sentiment analysis? I am not convinced by the short reasoning process. 2. Considering the conclusion is correct, then "Place the final layer at the front of the queue" is based on some prior knowledge about the task. Therefore, can I assume that without these inductive biases, the performance will be worse? Should I introduce the inductive bias for each downstream task?
6. It does not compare with some of the latest work. Like "Know what you don't need: Single-Shot Meta-Pruning for attention heads". Also, could it be combined with other works like compression? For example, your method then compression? Or does the implementation of your method hinder the implementation of other pruning/compression/quantization methods?
7. (Minor) Some tables are using figures. I think these tables can be generated via latex.


**Summary Of The Paper:**

This paper introduces a framework to prune parameters in transformer-based structures. The authors claim that the proposed method leads to a smaller model with faster and more accurate performance. It first replaces soft self-attention with hard self-attention by checking whether the replacement leads to a smaller loss and benefits more than N samples in the validation set. Using a similar way, it further prunes attention block, feed-forward block, and their neurons.


**Summary Of The Review:**

Overall, it is an interesting paper. An easy method to reduce the model size and increase the performance in the downstream tasks. Not sure whether it can be generalized to general transformer-based models. It looks more like some technology that is not so novel. There is also not enough detailed research analysis.

---

> ### Author Response · Authors · 2021-11-15
> **Response to Reviewer Hc1r (Part 1)**
>
> We thank the reviewer for their thoughtful review. We address their main concerns below.
>
> **Re: The pruning method is not very novel.**
>
> We agree that greedy pruning methods have been previously proposed. However, we would like to underscore that the novelty of our approach lies in the pruning objective. Previous pruning methods prune elements that have minimal impact on the model output (in order to improve efficiency with minimal loss in accuracy). This approach fundamentally cannot achieve accuracy improvements. Our accuracy-driven pruning method instead prunes elements that have a detrimental effect on the output (larger the better). This, in turn, leads to models that are more accurate while also being smaller and faster, as opposed to the accuracy-efficiency trade-offs typically involved in other pruning methods. In fact, contrary to other pruning methods, accuracy-driven pruning can be broadly seen as a form of training, since the objective of accuracy-driven pruning -- minimizing loss on the (validation) dataset -- is exactly the same as the objective of training. When large, over-parameterized models are trained on very limited data, the constraints of accuracy-driven pruning -- (1) loss can be reduced only be setting weights to 0 (equivalent to pruning elements), and no other changes to weights are allowed, and (2) a majority of samples in the validation set must benefit from each weight update; reduction in loss is a necessary but insufficient condition for weights to get updated -- help it learn more effectively than SGD. As a result, we find that training with SGD on a subset of the training data, followed by accuracy-driven pruning on the remaining unseen data, produces models with better generalization performance than training with SGD on the entire dataset.
>
> In addition, previously proposed techniques are also complementary to our method, and we demonstrate this in the sub-section of our results section titled “Previously proposed techniques to improve inference efficiency are more effective when applied to Specialized models”. Here, we show (see Figure 2 on page 7) that conventional pruning can be applied after Specialization to produce models that are even more efficient for a given accuracy constraint.
>
> **Re: In order to reduce the effect of outliers, the exact value of N is tuned for the different datasets, based on the validation loss of each sample." Could you introduce the details of tuning N?**
>
> We determine N as the number of elements in the validation dataset whose loss is less than the average loss of the misclassified samples. We consider samples whose loss is greater than the average loss of the misclassified samples to be outliers. We empirically validate this method to determine N on GLUE, and find that discarding outliers leads to significantly higher validation accuracy after Specialization. We also empirically verify that our metric to identify outliers leads to consistently highest validation accuracies on the GLUE tasks after Specialization. We have added this explanation to the paper in Section “Accuracy-driven pruning”, page 3.
>
> **Re: Not enough ablation study. Not sure what really matters. The pruning of which element matters the most?**
>
> In terms of accuracy, we find that there is no single element that, when pruned, greatly increases the generalization ability of the model. Significant accuracy gains are achieved only by pruning a combination of elements at multiple levels of granularity. Since the goal of accuracy-driven pruning is to find a model with minimum loss on the validation set, pruning an element from the model can be viewed as a step in the right direction towards a more accurate model with better generalization ability. This can be seen in Figure 3, where all elements are pruned in significant quantities in order to obtain the best models for the different downstream task, showing that there is no bias towards pruning a certain element.
>
> In terms of efficiency, we find that when small context lengths (<=512) are used, pruning FFN blocks leads to maximum speedups. At large context lengths, ATTN blocks become more time-consuming than FFN blocks, due to the quadratic complexity of self-attention. Therefore, pruning ATTN blocks provides the greatest speedups.

---

> > ### Author Response · Authors · 2021-11-15
> > **Response to Reviewer Hc1r (Part 2)**
> >
> > **Re: Lack mechanism analysis. For example, how does the attention to each word change after the pruning of each element?**
> >
> > Pruning just a single element does not have a significant impact on attention maps. However, we see significant changes in attention patterns when a set of elements are pruned together during Specialization. We have added a visualization in attention patterns before and after Specialization (Figure 5 in Appendix A.4.2), and this demonstrates the key reason for generalization performance gains from Specialization: the pre-trained model contains irrelevant information for the task at hand, which adds noise and ends up confusing the model, thereby hindering performance on the downstream task. Specialization helps the model focus on the task at hand, and hence improves performance on the downstream task by building meaningful word relationships.
> >
> > **Re: "For example, sentiment analysis requires only local context…”. How do you get this conclusion about the final layers and sentiment analysis? Can I assume that without these inductive biases, the performance will be worse? Should I introduce the inductive bias for each downstream task?**
> >
> > Yes, if elements are inspected in a random order, then performance will be worse (see Table 4). If we have no prior knowledge about the task, we divide the Transformer into three main functional regions: bottom layers (phrase-level information), middle layers (semantic and syntactic knowledge) and top layers (long-range dependency information). We consider all possible (3! = 6) orderings of the regions for pruning (bottom->middle->top, bottom->top->middle, middle->bottom->top, middle->top->bottom, top->bottom->middle and top->middle->bottom). Then, the Specialized model with minimum validation loss is chosen. We also find that randomly ordering elements within these functional regions has negligible accuracy on quality of the final model. For example, for sentiment analysis, we need to inspect top layers first to produce the most accurate models. However, inspecting the pre-final layer before the final layer (or vice-versa) has little impact on accuracy of the final model, further demonstrating the demarcation of linguistic knowledge in Transformer layers.
> > The use of inductive biases (from human intuition about the types of knowledge required for each task) only helps speed up the process by eliminating this search of region ordering, thereby reducing the overheads of Specialization without compromising on the quality of the final model.
> >
> > We arrive at our conclusions about sentiment analysis by validating our intuition with experimental results. We inspect the elements in different orders described above, and find that top->middle->bottom ordering leads to the most accurate model, which agrees with our intuition that sentiment analysis requires only local context (long-range information often ends up confusing the model, since sentiments often change rapidly), and that it is unlikely that syntactic and semantic information are needed. We thank the reviewer for pointing out the confusion, and have added this explanation to Appendix A.2 and A.3 to help validate our claims.
> >
> > **Re: It does not compare with some of the latest work. Like "Know what you don't need: Single-Shot Meta-Pruning for attention heads".**
> >
> > “Single-Shot Meta-Pruning for attention heads” performs task-agnostic pruning of attention heads to improve both fine-tuning and inference speed.  We believe our techniques are complementary, and can be applied after fine-tuning for a specific downstream task to improve accuracy and further improve efficiency. In the paper, we demonstrate that Specialization can be used along with DistilBERT and Q8BERT (Table 1 in page 6), two other task-agnostic optimization techniques, to improve both accuracy and efficiency. We thank the reviewer for pointing out this missing reference, and we have added it to the “related works” section.

---

> > > ### Author Response · Authors · 2021-11-15
> > > **Response to Reviewer Hc1r (Part 3)**
> > >
> > > **Re: Could this work be combined with other works like compression? For example, your method then compression? Or does the implementation of your method hinder the implementation of other pruning/compression/quantization methods?**
> > >
> > > Yes, this work can indeed be combined with other works like pruning/compression/quantization. We demonstrate this with results on DistilBERT (compression through knowledge distillation) in Table 1, Q8BERT (8-bit integer quantization) in Table 1 and Lottery Ticket Hypothesis which performs conventional pruning after Specialization in Figure 2. In all cases, we show that these techniques can be combined with Specialization to create accurate and highly efficient models. In particular, since our method is designed to identify and prune elements that have a detrimental impact on the output, we expect that all previously proposed compression methods that aim to minimize impact on output can be combined with Specialization.
> > >
> > >
> > > **Re: Not sure whether it can be generalized to general transformer-based models.**
> > >
> > > To the best of our knowledge, Specialization can be applied in a plug-and-play manner to any Transformer-based model while fine-tuning for any downstream task. We also present results on different kinds of architectures, including bidirectional and auto-regressive models.
> > >
> > >
> > > We hope we have addressed the main concerns. If not, please let us know and we will be happy to address them in subsequent responses and revisions.

---

> ### Author Response · Authors · 2021-11-27
> **Follow-up**
>
> We appreciate your insightful comments about our work. Since the discussion period is almost over, we were wondering if our rebuttal addressed your concerns. If not, please let us know and we will be happy to address them in subsequent responses.

---

> > ### Comment · Reviewer_Hc1r · 2021-11-30
> > **Reply**
> >
> > Your answers indeed help me to better understand your paper. However, main concerns still stand such as using prior knowledge and generalization problem. I will keep my score.

---

> > > ### Author Response · Authors · 2021-11-30
> > > **Use of prior knowledge**
> > >
> > > Thanks for the response! We would like to underscore that the use of prior knowledge about the task is only a heuristic to help speed up the Specialization process. **All the exact results in our paper can be obtained without the use of this prior knowledge.** We divide the Transformer into three main functional regions: bottom layers (phrase-level information), middle layers (semantic and syntactic knowledge) and top layers (long-range dependency information). We consider all possible (3! = 6) orderings of the regions for pruning (bottom->middle->top, bottom->top->middle, middle->bottom->top, middle->top->bottom, top->bottom->middle and top->middle->bottom). Then, the Specialized model with minimum validation loss is chosen. We also find that randomly ordering elements within these functional regions has negligible accuracy on quality of the final model (see Appendix A.2), and **this assumes absolutely no prior knowledge about the task**. Even with this exhaustive search, the time for (fine-tuning on a subset of the training data + Specialization on the unseen data) is no more than 15% larger than the time for fine-tuning on the entire dataset in all our experiments. With the use of prior knowledge, this reduces to 5% more than conventional fine-tuning. We again thank the reviewer for raising this concern. We will move the technique of searching functional regions of the Transformer (that uses no prior knowledge and leads to the exact same models as when prior knowledge is used) into the main paper, and the prior-knowledge based heuristic into the appendix in the final version. We hope this addresses your concerns on the generalizability of our techniques to any downstream task. We are happy to address any further concerns you may have also.

---

> > > > ### Comment · Area_Chair_qtyJ · 2021-12-01
> > > > **final thoughts?**
> > > >
> > > > Thanks for this discussion! Reviewer Hc1r, any further thoughts after reading the authors' responses to your review?

---

### Official Review · Reviewer_ccHS · 2021-11-03

**Correctness:** 3
**Technical Novelty And Significance:** 3
**Empirical Novelty And Significance:** 3
**Recommendation:** 8
**Confidence:** 3

**Main Review:**

Strength:
1. The proposed method is interesting, and it can be adopted by different models with significant improvement.
2. The accuracy-driven pruning is reasonable and different from previous pruning-based methods.
3. Leveraging hard attention is an interesting idea.
4. The paper is easy to read and the experiments are solid. The authors compare the method with multiple pruning methods including regularization and lottery ticket based methods. And the experiment shows that the proposed method Is better.

Weakness:
1. It would be better to have some more analysis on hard attention, such as some statistical comparison between soft-attention and hard-attention. Also, more analysis on training and inference speed between two attention mechanisms would be helpful.


**Summary Of The Paper:**

The authors propose a Specialization framework to create optimized transformer models for a given downstream task. The framework systematically uses accuracy-driven pruning. The authors proposed two ways to reduce model parameters, 1) Hierarchical pruning. Start from analyzing entire feed-forward and self-attention blocks, and inspect them at finer granularity (attention heads and neurons) only
when required. 2) Replacing soft-attention with hard-attention. The proposed method significantly improves benchmark models, BERT, Q8BERT and DistillBERT.

**Summary Of The Review:**

Overall, the proposed method is novel and reasonable. The experiments are quite solid by adopting the method to multiple structures and comparing it with SOTA pruning methods.

---

> ### Author Response · Authors · 2021-11-15
> **Response to Reviewer ccHS**
>
> We thank the reviewer for their thoughtful review. We address their main concerns below.
>
> **Re: It would be better to have some more analysis on hard attention, such as some statistical comparison between soft-attention and hard-attention.**
>
> We have added visualizations of class boundaries using T-distributed stochastic neighbor embedding to analyze how the principal components of final layer embeddings change with hard attention (Figure 4 in Appendix A.4.1). We have also added a visualization of how attention patterns in the final layer (word relationships) change with hard attention (Figure 5 in Appendix A.4.2).
>
> **Re: Also, more analysis on training and inference speed between two attention mechanisms would be helpful.**
>
> Hard attention is not used during training or fine-tuning, since it is not differentiable. After fine-tuning with vanilla soft attention, we selectively replace soft attention with hard attention in certain layers. Therefore, the use of hard attention does not affect the training or fine-tuning speed.
> Hard attention does not provide speedups on GPUs. This is because hard attention introduces unstructured sparsity, which the GPU platform used in our experiments cannot exploit. The primary benefit of using hard attention on GPUs is improvement in accuracy on the downstream task (shown in Table 4). We believe that hard attention can provide significant speedups on custom hardware accelerators designed to exploit unstructured sparsity (such as SCNN (Parasher et.al., 2017), SparTen (Gondimella et.al., 2019), etc.).
>
> We hope we have addressed the main concerns. If not, please let us know and we will be happy to address them in subsequent responses and revisions.

---

### Decision · Program_Chairs · 2022-01-20

**Decision:**

Reject

**Comment:**

The authors propose a simple and effective technique for task-specific pruning of transformer models that identifies which model components to prune by minimizing validation loss. Weaknesses of the paper include (1) related work reads more like a list and doesn’t compare and contrast the proposed approach with related work, (2) authors don’t compare to other structured pruning methods (that use different objectives) (3) lack of novelty — main difference with existing work is using validation loss to optimize and (4) one reviewer was unconvinced that the results should be possible given the approach. I share these concerns, and, in particular, I think they might be related. Given that the models are pruned using the development set (essentially equivalent to training on the development set), it seems infeasible that this approach could have been developed without looking at the testing data, and I’m concerned that this explains the unprecedentedly high accuracy compared to previous pruning approaches. At the very least, comparing to a baseline that trains on development data would be prudent in order to understand the result.